# BIRC5 Expression Is Regulated in Uterine Epithelium during the Estrous Cycle

**DOI:** 10.3390/genes11030282

**Published:** 2020-03-06

**Authors:** Minha Cho, Ok-Hee Lee, Eun Mi Chang, Sujin Lee, Sohyeon Moon, Jihyun Lee, Haeun Park, Keun-Hong Park, Chankyu Park, Kwonho Hong, Youngsok Choi

**Affiliations:** 1Fertility Center of Cha Gangnam Medical Center, Seoul 06135, Korea; mhcho@chauniv.ac.kr (M.C.); emchang@cha.ac.kr (E.M.C.); 2Department of Biomedical Science, CHA University, Gyeonggi-do 13488, Korea; okhlee@chamc.co.kr (O.-H.L.); sj091895@naver.com (S.L.); pkh0410@cha.ac.kr (K.-H.P.); 3Department of Stem Cell and Regenerative Biotechnology, Konkuk University, Seoul 05029, Korea; 1004sh.moon@gmail.com (S.M.); lsw46340@naver.com (J.L.); phe325@naver.com (H.P.); chankyu@konkuk.ac.kr (C.P.); hongk@konkuk.ac.kr (K.H.)

**Keywords:** *Birc5*, uterus, estrogen, progesterone

## Abstract

Baculoviral inhibitor of apoptosis repeat-containing 5 (*Birc5*), also known as survivin, is a member of the inhibitor of apoptosis (IAP) family of proteins and regulates the size of tissues through cell division control. The uterus is the most dynamically sized organ among tissues during the estrous cycle. Although *Birc5* is expressed in some terminally differentiated cells, the regulation of its expression in the uterus remains unknown. We investigated the regulation of *Birc5* expression in the mouse uterus. RT-PCR analysis showed that *Birc5* was expressed in various tissues, including the uterus; the expression level of *Birc5* was significantly higher at the diestrus stage. Immunohistochemistry and Western blotting analysis revealed that *Birc5* was more active in luminal and glandular epithelium than in endometrial stroma. In ovariectomized mice, *Birc5* expression in the uterus was gradually increased by estrogen treatment; however, progesterone injection decreased its expression. Estrogen-induced *Birc5* expression was blocked by treatment with estrogen receptor antagonist, ICI 182, 780 and progesterone-reduced *Birc5* expression was inhibited by the progesterone receptor antagonist RU486. These results suggest that *Birc5* expression is dynamically regulated by a combination of estrogen and progesterone via their receptor-mediated signaling.

## 1. Introduction

The uterus is a female reproductive organ that plays an important role in several pregnancy processes such fertilization, implantation, and embryonic development [1]. The endometrium of the uterus is periodically regulated according to the estrous cycle. In humans, the menstrual cycle has an interval of ≈28–30 days. It corresponds to the estrous cycle of 4–5 days in the rodents (reviewed in [2]). The estrous cycle consists of four stages viz. diestrus, proestrus, estrus, and metestrus. The precise regulation of the estrous cycle within ≈4–5 days is very important for a successful pregnancy. The uterine endometrium is dynamically regulated by the ovarian sex hormones estrogen and progesterone. The endometrium undergoes proliferation, differentiation, and shedding (humans and primates) or apoptosis (rodent) in response to hormones during the cycle [2].

Baculoviral inhibitor of apoptosis repeat-containing 5, (*Birc5*, also known as survivin) is an inhibitor of apoptosis (IAP). *Birc5* was originally identified and characterized in human cancers [3]. *Birc5* was highly expressed in various cancers such as lung cancer, ovarian cancer, breast cancer, brain tumor, colon cancer, pancreatic cancer, osteosarcoma, and cervical cancer (reviewed in [4]). It was used as a prognostic or survival marker in cancer patients. *Birc5* acts as inhibitor of apoptotic processes by repressing caspase activity via binding to caspase-3 and -7 in cancer cells, leading to the survival of cancer cells during tumorigenesis [5].

Recently, many studies o Bir5 expression in various tissue have been reported [4]. The expression of *Birc5* has been detected in various normal tissues such as the liver [6], arterial muscle [7], stomach [8], brain [9], ovary [10], testes [11,12,13], and uterus [1], though at lower levels than in cancers. *Birc5* has various functions in cellular processes such as differentiation and proliferation of stem cells and progenitor cells and its deficiency results in embryonic lethality at early stage of embryogenesis, viz. at embryonic day E4.5 [14]. This implies that *Birc5* plays a crucial role in cell differentiation and proliferation. In fact, T cell-specific *Birc5*-knocked-out mice showed that it plays a role in T cell development and differentiation [15,16]. Further, *Birc5* is involved in regulating proliferation of hematopoietic stem cells [17] and mesenchymal stromal cells [18]. Moreover, *Birc5* expression is regulated by various factors including p53 [19], PTEN [20], SIRT1 [21], HDAC2, and HDAC5 [22].

Almost 20 years ago, Konno and colleagues showed *Birc5* expression in normal uterus for the first time [1]; there was no further report in this regard about the uterus until recently. A recent study demonstrated that *Birc5* is aberrantly expressed in patients with endometrial hyperplasia [23]. They reported that *Birc5* is highly detected in the endometrium of patients with endometrial hyperplasia. Endometrial hyperplasia is caused by abnormally excessive proliferation of endometrial cells in the uterus, which is related to high estrogen level and low progesterone level. However, the regulatory relationship between *Birc5* expression and hormones in the normal uterus remains unknown. Therefore, we investigated whether *Birc5* is expressed and regulated by hormones in mouse uterus.

## 2. Materials and Methods

### 2.1. Animals

All animal studies were performed using 6–7-week-old ICR mice provided by KOATECH (Pyeongtaek, Korea). Mice were housed under controlled temperature and light conditions with lights on for 12 h daily and fed ad libitum. Animal care and experimental procedures complied with the Guide for the Care and Use of Laboratory Animals and were approved by the Institutional Animal Care and Use Committee (IACUC) of CHA University.

### 2.2. Estrous Cycle and Uterus Sampling

The stages of estrous cycle were determined by a vaginal smear as described in previous studies [24,25,26]. Approximately 0.2 mL of Dulbecco’s phosphate buffered saline (DPBS) was drawn into the pipette tip. The tip of the pipette was pushed gently into the entrance of the vagina at a depth of 2–5 mm, and the fluid was flushed into the vagina and then backed up into the pipette; this was repeated two to three times. The collected DPBS was dropped onto a glass slide and dried. Cell staining was carried out by hematoxylin and eosin. After incubating the cells in 50%, 75%, and 90% ethanol for 5 min each, they were stained with hematoxylin (Vector Laboratory, USA). The slides were washed in tap water for 3 min and soaked in eosin Y (Sigma-Aldrich, St. Louis, MO, USA) for 10 min. After the eosin treatment, the dyed slides were in turn dipped in 90% ethanol and then 100% ethanol for 5 min. After a short incubation in xylene, the slides were covered with cover slips using Permount mounting medium (Thermo Fisher Scientific, Waltham, MA, USA). The staining was observed with a microscope and the stage of the estrous cycles was determined by vaginal smear cytology [27]. Once the estrous cycle was correctly identified, the uterus was isolated. Half of the uterus was processed for RNA and protein extraction and the other half was used for paraffin block.

### 2.3. Ovariectomy and Hormone Treatments

To examine the effects of hormones on *Birc5* expression in the uterus, 7-week-old ICR mice were ovariectomized (OVX). After a stabilization period of 2 weeks, the ovariectomized mice were subcutaneously injected with E_2_ (200 ng/mouse, Sigma-Aldrich, St. Louis, MO, USA) and/or P_4_ (2 mg/mouse, Sigma-Aldrich, St. Louis, MO, USA). Mice were sacrificed by cervical dislocation and uteri were collected at 0, 3, 6, and 24 h after E_2_ or/and P_4_ treatment. To investigate whether the expression of *Birc5* in uterus is dependent on E_2_ or/and P_4_, OVX mice were injected with an ER antagonist, ICI 182,780 (500 μg/mouse, Sigma-Aldrich, St. Louis, MO, USA) or PR antagonist, RU486 (1 mg/mouse) 30 min before E_2_ or P_4_ injection. Sesame oil (100 μL/mouse, Sigma-Aldrich, St. Louis, MO, USA) was used for the control mice.

### 2.4. RNA Preparation, Reverse Transcription PCR (RT-PCR), and Quantitative Real-Time PCR (qRT-PCR)

The tissues isolated from the mice were stored in RNA later solution (Sigma-Aldrich, St. Louis, MO, USA) overnight at 4 ℃. After removing RNA later solution, the tissue was homogenized in Trizol reagent (Invitrogen, Waltham, MA, USA) using a homogenizer and then purified with chloroform (Sigma-Aldrich, St. Louis, MO, USA). Total RNA (1 μg) was reverse transcribed using SensiFast™ cDNA Synthesis Kit (Bioline, London, UK) according to the manufacturer’s protocol. The product was used for reverse transcription PCR (RT-PCR) and quantitative real-time PCR (qRT-PCR) analyses. RT-PCRs were carried out using C1000 Thermal Cycler (Bio-Rad, Hercules, CA, USA). PCR conditions and size of gene-specific primers are shown as Appendix A. The PCR products stained with Loading Star (Dyne Bio, Seoul, Korea) were analyzed by gel electrophoresis on 2% agarose gel using Chemidoc™ XRS+ system (Bio-Rad, Hercules, CA, USA). Quantitative RT-PCR (qRT-PCR) was performed by the CFX96 Touch (Bio-Rad, Hercules, CA, USA) using iQ™ SYBR Green Super Mix (Bio-Rad, Hercules, CA, USA). qRT-PCR conditions were as follows: 40 cycles of denaturation at 95 ℃ for 15 s, annealing at 63 ℃ for 15 s, and extension at 72 ℃ for 20 s followed by 10 min at 72 ℃, 10 s at 95 ℃, and 5 s at 65 ℃. To quantify the level of gene expression, the cycle threshold (C_T_) value was calculated. The C_T_ value for each gene was determined in the linear phase of the amplification and normalized to the C_T_ value of either ribosomal protein L-7 (Rpl7) or Gapdh to obtain the ΔΔC_T_. The 2^−ΔΔCT^ method was used for obtaining the fold change of each gene [28].

### 2.5. Immunohistochemistry

Uteri collected from mice were fixed for 1 week in 4% paraformaldehyde (PFA) at 4 ℃ in a paraffin block. The embedded uteri were sectioned at 5 µm thickness using a paraffin microtome (Macroteck, Ilsan-si, Korea) and put on adhesion microscope slides (Paul Marienfeld, Lauda-Konigshofen, Germany). Sectioned uteri on the slide were deparaffinized twice in xylene (Duksan, Seoul, Korea) for 7 min and rehydrated by immersing once each in 100% ethanol (DaeJung, Seoul, Korea), 90% ethanol, 75% ethanol, and 50% ethanol for 5 min, followed by washing the slides once for 5 min under tap water. Slides were treated with 3% hydrogen peroxide for 10 min to block endogenous peroxide. Deparaffinized slides were boiled in antigen retrieval buffer (10 mM sodium citrate, 0.05% Tween 20, pH 6.0), cooled at room temperature (RT) for 1 h, and then washed in tap water for 5 min followed by three washes using phosphate buffered saline containing 0.05% Tween-20 (PBS-T). After removing PBS-T around tissue section using 3M paper, a hydrophobic barrier was drawn with ImmEdge hydrophobic pen (Vector Laboratories, USA) on the periphery of the tissue. Sections were blocked with PBS-T containing 4% bovine serum albumin (BSA) and 5% normal goat serum (Vector Laboratories, USA) for 1 h at RT in a humidified chamber. Blocking sections were incubated with anti-survivin mouse monoclonal IgG (1:500 dilution, NB500-238, Novus Biologicals, Centennial, CO, USA) in PBS-T with 4% BSA at 4 ℃ overnight. Slides were washed with PBS-T three times and subsequently were treated with secondary antibody viz. anti-mouse antibody conjugated with HRP (1:200 dilution, G21040, Sigma-Aldrich, St. Louis, MO, USA) for 1 h at RT. Normal mouse IgG was used as a negative control. Peroxide signals were developed using DAB substrate kit (SK-4100, Vector Labs, Burlingame, CA, USA) according to the manufacturer’s instruction. After washing, the slides were counter-stained with hematoxylin, washed in tap water, and dehydrated in 50% ethanol, 75% ethanol, 90% ethanol, 100% ethanol for 5 min each followed by two washes in xylene for 7 min each. IHC sections were mounted on cover slips with Permount Mounting Medium (StatLab, Lodi, CA, USA). Images were captured and analyzed by Primo Star (Zeiss, Oberkochen, Germany).

### 2.6. Western Blotting

Uteri from adult mice were collected and frozen in liquid nitrogen. The samples were added to 600 µL RIPA lysis buffer containing protease inhibition cocktail and grounded using a homogenizer (VWR International, Radnor, PA, USA) with vortexing every 10 min while being placed on ice for 30 min. After the incubation, the samples were centrifuged at 13,000 rpm for 30 min at 4 ℃ and the supernatant containing proteins was collected and stored at –80 ℃. Equal amounts of protein samples (10~20 μg) were separated by sodium dodecyl sulphate-polyacrylamide gel electrophoresis (SDS-PAGE) using a 10%–12% running gel and then transferred to polyvinylidene difluoride (PVDF) membranes (Bio-Rad, Hercules, CA, USA) for 1 h and 10 min. Membranes were washed in Tris-buffered saline containing 0.05% Tween-20 (TBS-T), blocked in ProNA™ phospho-block solution (TransLab, Dajeon-si, Korea) for 1 h at RT, and incubated with anti-survivin mouse monoclonal IgG (1:1000 dilution, NB500-238, Novus, Centennial, CO, USA) overnight at 4 ℃. After washing three times in TBS-T, the blots were incubated with HRP-conjugated goat anti-mouse antibody (1:5000 dilution, sc2005, Santa Cruz Biotechnology Inc., Dalls, TX, USA) in ProNA™ phospho-block solution for 1 h at RT. The membranes were washed three times using TBS-T, and subsequently, the blots were developed using ECL™ Prime Western Blotting Detection Reagent (GE Healthcare, Chicago, IL USA). To check whether the same amount of protein was loaded for each sample, anti-GAPDH antibody (1:2000 dilution, 14C10, Cell signaling technology, Danvers, MA, USA) was used as a loading control. The relative expression of the protein bands was quantified by Image Lab program.

### 2.7. Statistical Analysis

The experimental data were reported as mean ± standard error of mean (±SEM). The results were analyzed by one-way ANOVA for statistical evaluation. A *p*-value of less than 0.05 was considered as statistically significant.

## 3. Results

### 3.1. Expression of Birc5 in the Mouse Uterus

In order to confirm the expression of *Birc5* mRNA, we performed RT-PCR analysis using total RNAs, which were extracted from various tissues in the mouse (Figure 1a,b). As shown in Figure 1, *Birc5* expression was barely detected in the rest of the tissues including the kidney, liver, heart, brain, and lung (Figure 1a). These results are almost consistent with those from the previous studies indicating that *Birc5* expression is detected in normal tissues such as the liver [6], stomach [8], brain [9], ovary [10], and testes [11,12,13]. However, *Birc5* transcripts were detected in high quantities in several tissues including the small intestine, stomach, spleen, ovary, and uterus in the female mice (Figure 1b). After confirming the expression of *Birc5* mRNA in the mouse uterus, we examined the expression pattern of *Birc5* during the estrous cycle. Vaginal smear assay was used to distinguish each stage of the estrous cycle viz. diestrus, proestrus, estrus, and metestrus (Figure 1c). The results of RT-PCR and qRT-PCR showed a cyclic change in *Birc5* expression during the estrous cycle (Figure 1d,e). *Birc5* expression level was the highest in diestrus stage and then reduced in the rest of the cycle including in the proestrus, estrus, and metestrus stages (Figure 1e).

### 3.2. Localization and Expression Level of BIRC5 Protein in Mouse Uterus during the Estrous Cycle

To investigate the localization and expression of BIRC protein in the mouse uterus, immunohistochemistry (IHC) was performed using mouse monoclonal *BIRC5* antibody in the uterus. IHC results showed that *BIRC5* was present in all layers of the endometrium (viz. luminal epithelium (LE), glandular epithelium (GE), stroma) (Figure 2a). Interestingly, strong staining was observed in the luminal epithelium and glandular epithelium compared to that in the endometrial stroma. Western blotting showed a relatively similar *BIRC5* expression level during the estrous cycle (Figure 2b,c). Results showed that *BIRC5* expression was slightly higher in the diestrus stage than that in metestrus (Figure 2c).

### 3.3. Regulation of Birc5 Expression by Estrogen Treatment

Previous results have indicated that *Birc5* transcript and its protein were dynamically regulated during the estrous cycle of the uterus. Cyclic modulation of the normal uterine endometrium is regulated by two ovarian steroid hormones, namely, estrogen and progesterone. Therefore, we first examined the effect of estrogen (E_2_) on the expression of *Birc5* using ovariectomized (OVX) mice. The OVX mice were treated with E_2_ and their uteri were obtained at 0, 3, 6, 12, and 24 h after E_2_ treatment. RT-PCR analysis indicated that the expression of *Birc5* mRNA gradually increased with time after E_2_ administration (Figure 3a). Early growth response 1 (*Egr1*) and Lactoferrin (*Ltf*) expressions were used as referral genes marking estrogen early and late response genes, respectively (Figure 3b). The gradual induction of *Birc5* expression by estrogen treatment was confirmed by semi-quantitative RT-PCR. The relative increase in *Birc5* expression peaked at 24 h after E_2_ treatment (Figure 3b). Next, we examined the status of *BIRC5* protein in uterine endometrium of OVX mice by immunostaining with *BIRC5* antibody. IHC results showed that *BIRC5* protein was weakly detected without E_2_ treatment (0 h) and was heavily stained for in all layers of the uterus over time (Figure 3c). Unlike the RT-PCR results, the signal for *BIRC5* was the strongest at 6 h post E_2_ injection. This suggests that E_2_ promotes increase in *Birc5* expression in the uterus during the estrous cycle.

### 3.4. Regulation of Birc5 Expression by Progesterone Treatment

Next, we analyzed whether progesterone (P_4_) affects the expression of *Birc5* in the uterus. As in the previous studies, we used ovariectomized (OVX) female mice. The OVX mice were injected with P_4_ (2 mg/mouse), and the uterus samples were collected at 0, 3, 6, and 24 h. Interestingly, P_4_ treatment in OVX mice showed no significant effect on *Birc5* expression unlike estrogen treatment. (Figure 4a,b). Amphiregulin (*Areg*) and Homeobox *A10* (*Hoxa10*) were used to confirm that the reaction with progesterone occurred properly (Figure 4a). qRT-PCR analysis showed that the expression level was lowest at 6 h after P_4_ treatment, but it was not a significant difference (Figure 4b).

IHC analysis also showed that *BIRC5* protein was detected as basal level in the uterine endometrium of OVX mice without P_4_ treatment. However, its signal in the uterine endometrium of OVX mice was rapidly reduced after the administration of P_4_ (Figure 4c). This indicates that P_4_ regulates the time difference in *Birc5* expression in the uterus during the estrous cycle. In fact, P_4_ like E_2_, is secreted alternately from the ovary during the estrous cycle and acts jointly or solely on endometrial changes.

### 3.5. Hormone-Dependent Regulation of Birc5 Expression via Hormone Receptors in the Uterus

The expression of *Birc5* rapidly responded to hormones E_2_ and P_4_ in the uterine endometrium (Figure 3 and Figure 4). This rapid regulation of target gene expression by hormones occurs via their receptors. Therefore, we investigated whether *Birc5* expression is regulated via the estrogen receptor (ER) and progesterone receptor (PR). First, an ER antagonist, ICI 182,780 (ICI), was used to determine whether ER was involved in the regulation of *Birc5* expression. ICI was injected in OVX mice for 30 min before E_2_ administration. *Birc5* expression was examined in the absence or the presence of E_2_ administration (24 h). As shown in Figure 5a, the relative expression of *Birc5* mRNA was dramatically increased by E_2_ treatment. However, its induction was prevented with ICI pretreatment in OVX mice (Figure 5b).

Next, a PR antagonist, RU486, was used to examine whether PR was involved in the downregulation of *Birc5* expression. In addition, we looked at the action of PR on ER-dependent increase in *Birc5* expression in the uterus. As shown in Figure 5c,d, the increase in *Birc5* expression by E_2_ treatment (6 h) was significantly reduced by co-treatment with E_2_ and P_4_ (Figure 5d). This effect of P_4_ on the estrogen-dependent increase was reversed by the treatment with PR antagonist, RU486 (Figure 5d). *BIRC5* protein was detected at higher level in E_2_-treated uterine endometrium compared with the sesame oil-treated group. The effective induction by E_2_ was reduced by progesterone, whereas this effect was again reversed while using P_4_ antagonist, RU486, in the presence of estrogen in the uterine endometrium (Figure 5e). These results indicate that *Birc5* expression in the uterus is precisely and dynamically regulated by E_2_ and P_4_.

## 4. Discussion

In this report, we presented the regulation of *Birc5* expression in the uterus. We found that *Birc5* is periodically expressed in the uterus during the estrous cycle (Figure 1 and Figure 2). Moreover, we showed that its cyclic expression is related to two ovarian hormones, namely, estrogen and progesterone (Figure 3 and Figure 4). Interestingly, we demonstrated that estrogen positively regulates *Birc5* expression through its receptor in the uterine epithelium (Figure 5). This needs further study to prove the involvement in *Birc5* expression. This implies that *Birc5* plays a role in the uterine dynamics during the estrous cycle in normal status.

As a member of the IAP family, *Birc5* is expressed in various cells including hematopoietic cells, immune cells and vascular endothelial cells, and mesenchymal stromal cells [4,17,18]. Additionally, *Birc5* plays a role in regulating cell proliferation and survival. One of its functions is to inhibit apoptosis by acting on caspases in the cytoplasm and mitochondria. Further, *Birc5* is also located in the nucleus and regulates cell cycle factors [29,30].

The uterus, which is composed of endometrium and myometrium, is a part of the female reproductive system. The uterine endometrium consists of a single columnal epithelium and a connective tissue stromal layer. The uterine layers are cyclically changed with periodic proliferation, differentiation, and apoptosis. This cycle is called the estrous cycle in the rodents. The remodeling of the uterus is controlled by female sex steroid hormones such as estrogen (E_2_) and progesterone (P_4_). Therefore, we investigated the relationship between *Birc5* expression and uterine endometrium.

Nabilsi et al. have reported the relationship between *BIRC5* and sex hormones in the human uterus [31]. They analyzed the expression of *BIRC5* in proliferative and secretory phases of normal menstrual cycle and found that *BIRC5* expression was higher in the proliferative phase than in the secretory phase. They further observed that *BIRC5* expression was increased in the human endometrium after estrogen treatment, which then decreased upon progestin treatment [31]. It was the first evidence showing that *BIRC5* expression is regulated by female sex hormones. However, it remained unclear whether the hormones acted alone or together.

In fact, although progesterone and estrogen secretion stages are different, they are organically associated rather than acting separately. Skinner et al. reported that progesterone is critical for the full effect of estrogen in ewes [32]. Therefore, we used the OVX model to show their individual effect on *BIRC5* expression in the uterus. Each treatment on OVX mice showed a similar pattern as that observed in the study by Nabilsi and colleague. *Birc5* transcripts were increased with E_2_ administration into OVX mice and this increase was significantly reduced by pretreatment with ER antagonist ICI 182,780. This suggests that *Birc5* is regulated in the uterine epithelium via estrogen receptor-associated hormonal signaling. It has been shown that *Birc5* is regulated by various signaling pathways such as PI3K/Akt, mTOR, ERK, and NFκB pathways [33,34]. Interestingly, the increased expression of *Birc5* was decreased by P_4_ administration. This suggests that *Birc5* is regulated by both E_2_ and P_4_, during the dynamic uterine cycle.

In conclusion, the present study demonstrated that *Birc5* may be periodically regulated by estrogen and progesterone in the endometrium of the uterus during the estrous cycle. This finding provides insights into understanding the regulatory mechanism underlying the endometrial dynamics.

## Figures and Tables

**Figure 1 genes-11-00282-f001:**
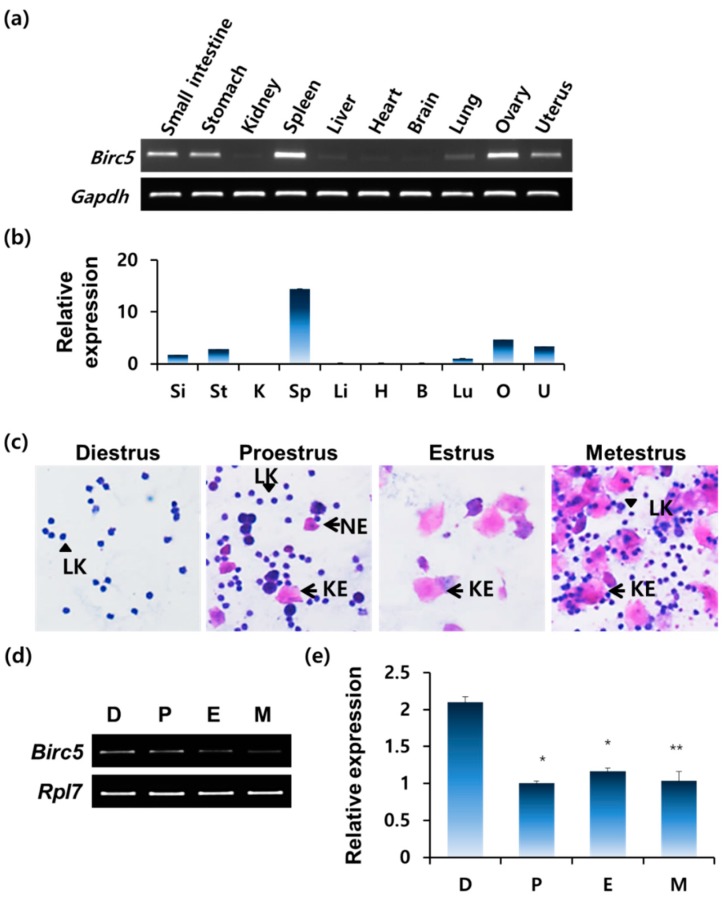
Baculoviral inhibitor of apoptosis repeat-containing 5 (*Birc5*) expression in mouse tissues. (**a**) RT-PCR analysis of *Birc5* mRNA in the small intestine, stomach, kidney, spleen, liver, heart, brain, lung, ovary, and uterus using total RNA extracted from a 6-week-old female mouse (*n* = 4). *Gapdh* transcript was used as an internal control. (**b**) Quantitation of the relative mRNA expression of *Birc5* in mouse tissues: Si, small intestine; St, stomach; K, kidney; Sp, spleen; Li, liver; H, heart; B, brain; Lu, lung; O, ovary; U, uterus. The criterion of relative expression change was based on the value at small intestine. (**c**) Vaginal smear assay to classify the estrous cycle: NE, nucleated epithelial cell; LK, leukocyte; KE, keratinized epithelial cell. (**d**) RT-PCR results of *Birc5* expression in mouse uterus at different estrous stages (*n* = 4) confirmed by vaginal smear assay. D, diestrus; P, proestrus; E, estrus; M, metestrus. (**e**) Quantitation of the relative mRNA expression of *Birc5* in mouse uterus. The criterion of relative expression change was based on the value at metestrus stage of the estrous cycle. Expression levels were normalized against *Rpl7* mRNA. D, diestrus; P, proestrus; E, estrus; M, metestrus. * *p*-value < 0.05, ** *p*-value < 0.01.

**Figure 2 genes-11-00282-f002:**
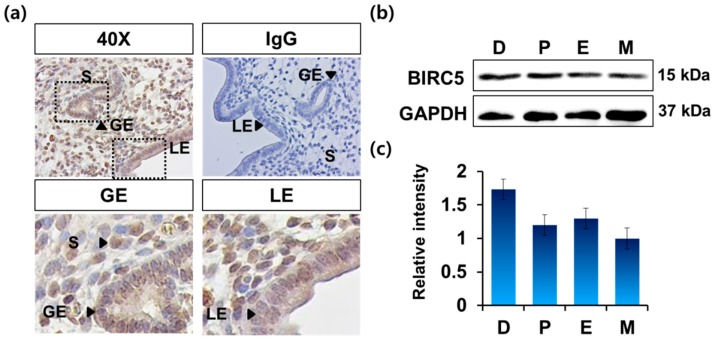
Expression of *BIRC5* protein in mouse uterus during the estrous cycle. (**a**) Immunohistochemical analysis of *BIRC5* in mouse uterus (*n* = 4). LE, luminal epithelium; GE, glandular epithelium; S, stroma. (**b**) Western blotting analysis of *BIRC5* using whole cell lysate of mouse uteri from each stage of the estrous cycle (*n* = 4). GAPDH antibody (GAPDH) was used for internal control. D, diestrus; P, proestrus; E, estrus; M, metestrus. (**c**) Quantitation of the relative levels of *BIRC5* in uterus at each stage during the estrous cycle using Image Lab program. The criterion of relative expression change was based on the value at the metestrus stage of the estrous cycle. D, diestrus; P, proestrus; E, estrus; M, metestrus.

**Figure 3 genes-11-00282-f003:**
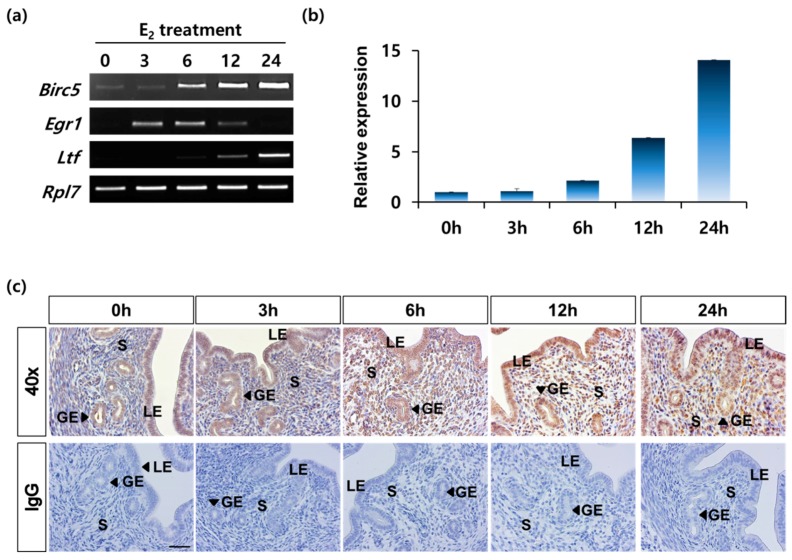
*BIRC5* expression induced by estrogen treatment in ovariectomized (OVX) mice. (**a**) RT-PCR result in the uterus of the OVX mouse after E_2_ treatment for 0, 3, 6, 12, and 24 h (each group *n* = 4). Early growth response 1 (*Egr1*) and Lactoferrin (*Ltf*) genes were used to confirm the appropriate E_2_ response. Ribosomal protein L7 (*Rpl7*) gene was used for internal control. (**b**) qRT-PCR analysis showed the relative fold changes in *Birc5* expression caused by E_2_ treatment. The criterion of relative expression change was based on the value at 0 h after E_2_ treatment. (**c**) Localization of *BIRC5* in the uterus of OVX mice treated with E_2_. Normal mouse IgG was used as negative control. LE, luminal epithelium; GE, glandular epithelium; S, stroma. Scale bar, 100 μm.

**Figure 4 genes-11-00282-f004:**
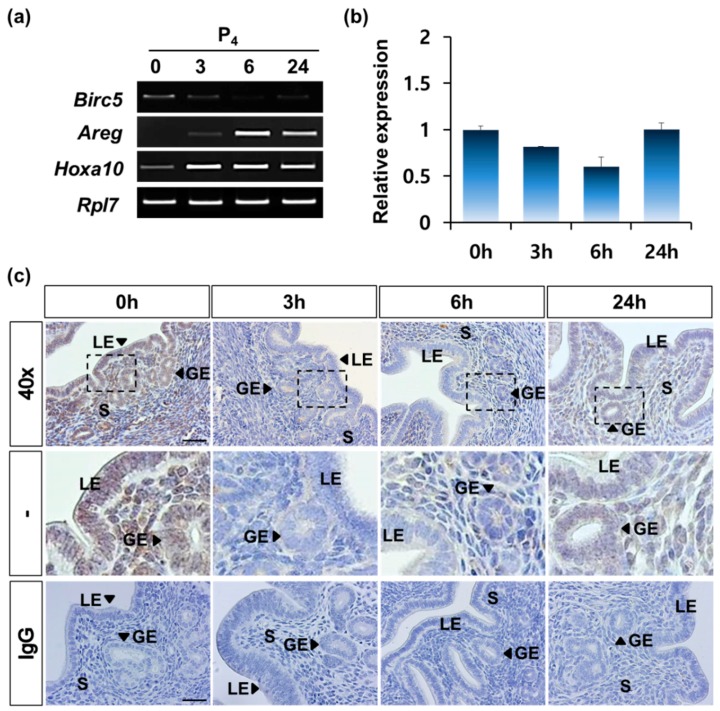
Reduction in *Birc5* expression by progesterone in OVX mice. (**a**) RT-PCR analysis of *Birc5* expression in the uterus of OVX mouse after P_4_ treatment for 0, 3, 6, and 24 h (each group *n* = 4). Amphiregulin (*Areg*) and Homeobox A10 (*Hoxa10*) were used to confirm the appropriate P_4_ response. Ribosomal protein L7 (*Rpl7*) gene was used as an internal control. (**b**) qRT-PCR result showed the relative fold changes in *Birc5* expression post P_4_ treatment. The criterion of relative expression change was based on the value at 0 h after E_2_ treatment. (**c**) Immunohistochemical analysis of *BIRC5* in the uterus of OVX mouse treated with P_4_. IgG was used as negative control. Scale bar, 100 μm.

**Figure 5 genes-11-00282-f005:**
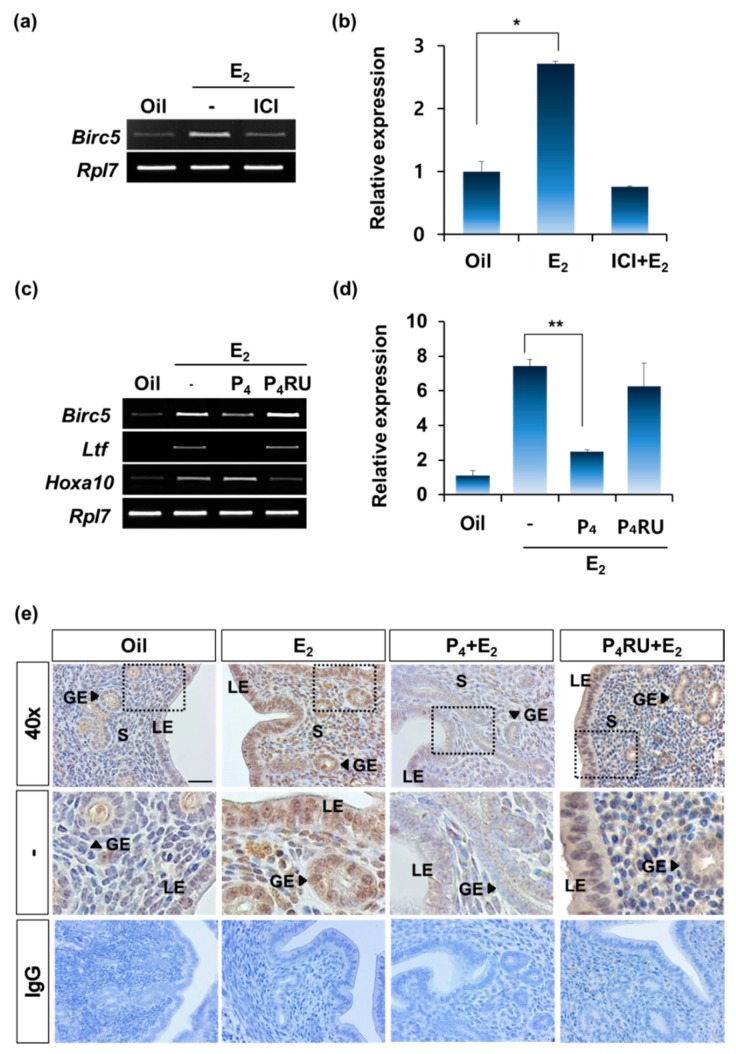
Regulation of *Birc5* expression via estrogen and progesterone receptors. (**a**) Estrogen receptor (ER) antagonist, ICI 182,780 (ICI) was applied before E_2_ treatment and the mice were sacrificed at 24 h after the E_2_ treatment. *Birc5* expression was analyzed by RT-PCR (each group *n* = 4). (**b**) qRT-PCR was used to quantify the relative level of *Birc5* expression. The criterion of relative expression change was based on the value at oil treatment in OVX mice. Sesame oil, Oil; ICI 182,780 (ICI); and estrogen, ICI+E_2_. *, *p*-value (*p* < 0.01). (**c**) RT-PCR analysis of *Birc5* expression in the uterus of OVX mice after treatment with E_2_ and co-treatment with E_2_ + P_4_ or RU486 + E_2_ + P_4_. (each group *n* = 4). Lactoferrin (*Ltf*) and Homeobox A10 (*Hoxa10*) were used to confirm the appropriate hormone response. Ribosomal protein L7 (*Rpl7*) gene was used for internal control. (**d**) qRT-PCR showed the relative fold changes in *Birc5* expression. The criterion of relative expression change was based on the value at oil treatment in OVX mice. **, *p*-value (*p* < 0.01). (**e**) Immunohistochemical analysis of *BIRC5* in uteri of OVX mice treated with E_2_, P_4_ + E_2_ or RU486 + E_2_ + P_4_ (P_4_RU + E_2_).

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
