# Peer review of "BIRC5 Expression Is Regulated in Uterine Epithelium during the Estrous Cycle"

_genes, 2020, doi:10.3390/genes11030282_

Round 1

Reviewer 1 Report

1- lines 15-29 and throughout the manuscript, authors need to clarify what makes BIRC5 gene interesting to study in uterus tissue.

2- line 22, please change “located” to “more active”.

3- lines 126-134, please include a sentence to explain why RT-PCR peak (24h after E2 treatment) is so different from protein results (6h after E2 treatment).

4- line 157, “this indicates that P4 either 158 negatively regulates or does not affect Birc5 expression in the uterus during the estrous cycle”. please rephrase this sentence as its not consistent with the rest of results, e.g. “The expression of Birc5 rapidly responded to hormones E2 and P4 in the uterine endometrium” (line 167) or “These results indicate that Birc5 expression in the uterus is precisely regulated by E2 and P4” (line 185).

Author Response

We thank for the reviewer's precious comments.

Please find an attached file showing our reponses to you.

Reviewer 2 Report

In the current study, Cho et al. have investigated the regulation of Birc5 expression in the mouse uterus during the estrous cycle. In addition, using inhibitors they show Birc5 expression is dynamically regulated by a combination of estrogen and progesterone via their receptor-mediated signaling. This study is important because it provides insights into understanding the basic regulatory mechanism of BIRC5 in the endometrium which could be extrapolated to pathophysiological conditions such as endometrial hyperplasia, endometriosis etc. Overall, this is a well-designed study depicting proper conclusion and interpretation. However, the study has several concerns that the authors need to address for the manuscript to be considered for publication.

  1. In lines 61-62, the authors state “A recent demonstrated that Birc5 is aberrantly expressed in endometrial hyperplasia [23]”. The authors need to specify whether BIRC5 was over- or under-expressed in this study? And mention the species (I am assuming it to be mouse).
  2. Line 77 “Especially, the uterus showed high Birc5 expression level” is redundant. Delete it.
  3. A general comment for all figures, report the animal number (N) in the figure legends. In figures showing histogram, the authors need to state the expression was relative to which group i.e., which group expression was arbitrarily set to 1.
  4. For figure 1b and 1e, please mention the relative expression was arbitrarily set to 1 for which group? I am assuming in the histogram for figure 1e, the relative expression was compared to proestrus (set as 1), and the other groups were accordingly valued. Or else the authors should state the y axis as per the ddCT. Also in the legend for figure 1b and 1e, clearly depict that they are qRT-PCR analysis. For figure 1c, make sure that abbreviation KE in proestrus and LK in diestrus is properly depicted.
  5. In line 102, the authors state that “Interestingly, none of the stromal cells were stained”. Similarly, in line 104, the authors state BIRC5 staining “rarely in the endometrial stroma”. However, in figure 2a inset of GE, it depicts BIRC5 staining in stromal cells. Please explain. Also, to what stage does the figure 2a IHC belong? For figure 2c if the relative expression was arbitrarily set to 1 for any group, then the authors need to mention it. If BIRC5 is negatively regulated by P4, why is BIRC5 high in diestrus when P4 levels are high? Also, the authors talk about BIRC5 expression using IHC analysis in endometrium, but what about myometrial BIRC5 expression? Because the western blots depict whole uterine BIRC5 expression which could be contributed by both endometrial and myometrial compartments. The authors clearly need to show myometrial BIRC5 expression and its contribution in the current study either using IHC or western blot. In such cases, generally endometrial layer is gently scraped off to get mostly myometrium free of endometrium and can be later used for immunoblot analysis.
  6. In line 201-202, the authors state that “Interestingly, we demonstrated that the hormones positively and cooperatively regulate Birc5 expression through their own receptors in the uterine epithelium (Fig. 5)”. This suggests that both hormones positively regulate BIRC5 expression. However, E2 positively regulates while P4 negatively regulates BIRC5 expression. Explain this discrepancy. There is no doubt that the Birc5 expression in the uterus is precisely regulated by E2 and P4. However, the mechanism of action is not described in this paper i.e., which receptors are specifically involved in BIRC5 regulation. Moreover, in line 170, the authors state that “Birc5 expression is regulated via the non-genomic action of hormone receptor such as estrogen receptor (ER) and progesterone receptor (PR)”. On what basis have the authors deducted that BIRC5 expression is solely regulated via non-genomic action of hormone receptors and is not regulated by genomic action of ESR1 and Pgr. Firstly, the authors need to demonstrate the localization of BIRC5 (whether it is cytoplasmic or nuclear). Secondly, whether nuclear or membrane receptors are involved in mediating RU486 and ICI action needs to be elaborated.
  7. The discussion section needs editing and revision. For example, sentence in lines 205-206 “As a member of the IAP family, most Birc5 including hematopoietic cells, immune cells and vascular endothelial cells, and mesenchymal stromal cells [4, 17, 18]” is incomplete and means nothing. Similarly, in line 215 “Therefore, we investigated the relationship” is incomplete means nothing.
  8. In line 269, correct “Sesame oil (100 λ/mouse, Sigma-Aldrich, USA)” to 100ul.
  9. For statistical analysis, the authors have used student’s unpaired t-test instead of 1-way ANOVA. For groups more than 2, the authors need to analyze data using 1-way ANOVA followed by posthoc analysis.

Author Response

We thank the reviewer's precious comments. Please find an attached file containing line-by-line response.

Round 2

Reviewer 2 Report

In the current study, Cho et al. have investigated the regulation of Birc5 expression in the mouse uterus during the estrous cycle. In addition, using inhibitors they show Birc5 expression is dynamically regulated by a combination of estrogen and progesterone via their receptor mediated signaling. This study is important because it provides insights into understanding the basic regulatory mechanism of BIRC5 in the endometrium which could be extrapolated to pathophysiological conditions such as endometrial hyperplasia, endometriosis etc. Overall, this is a well-designed study with sufficient number of experimental animals for statistical conclusion and interpretation. The authors have improved the quality of the manuscript upon revision which can now be considered for publication.